# Resistance to Trastuzumab

**DOI:** 10.3390/cancers14205115

**Published:** 2022-10-19

**Authors:** Sneha Vivekanandhan, Keith L. Knutson

**Affiliations:** Department of Immunology, Mayo Clinic, 4500 San Pablo Rd., Jacksonville, FL 32224, USA

**Keywords:** herceptin, tyrosine kinase inhibitor, vaccine, immune responses, monoclonal antibody, antibody drug conjugate, pertuzumab

## Abstract

**Simple Summary:**

Trastuzumab is a humanized antibody that has significantly improved the management and treatment outcomes of patients with cancers that overexpress HER2. Many research groups, both in academia and industry, have contributed towards understanding the various mechanisms engaged by trastuzumab to mediate its anti-tumor effects. Nevertheless, data from several clinical studies have indicated that a significant proportion of patients exhibit primary or acquired resistance to trastuzumab therapy. In this article, we discuss underlying mechanisms that contribute towards to resistance. Furthermore, we discuss the potential strategies to overcome some of the mechanisms of resistance to enhance the therapeutic efficacy of trastuzumab and other therapies based on it.

**Abstract:**

One of the most impactful biologics for the treatment of breast cancer is the humanized monoclonal antibody, trastuzumab, which specifically recognizes the HER2/neu (HER2) protein encoded by the *ERBB2* gene. Useful for both advanced and early breast cancers, trastuzumab has multiple mechanisms of action. Classical mechanisms attributed to trastuzumab action include cell cycle arrest, induction of apoptosis, and antibody-dependent cell-mediated cytotoxicity (ADCC). Recent studies have identified the role of the adaptive immune system in the clinical actions of trastuzumab. Despite the multiple mechanisms of action, many patients demonstrate resistance, primary or adaptive. Newly identified molecular and cellular mechanisms of trastuzumab resistance include induction of immune suppression, vascular mimicry, generation of breast cancer stem cells, deregulation of long non-coding RNAs, and metabolic escape. These newly identified mechanisms of resistance are discussed in detail in this review, particularly considering how they may lead to the development of well-rationalized, patient-tailored combinations that improve patient survival.

## 1. Introduction

Breast cancer is a leading cause of cancer-related deaths in women worldwide [1]. HER2 is part of a family of transmembrane receptors, it is overexpressed in about 15–20% of invasive breast cancers (BC), and it is associated with aggressive biology and a natural history of shortened survival [2]. Trastuzumab, a humanized monoclonal antibody developed initially for breast cancer, specifically recognizes the HER2 protein encoded by the *ERBB2* gene. Trastuzumab has had a wide-ranging impact on the management of patients with HER2+ advanced and early BC, as well as other cancers. The FDA originally approved trastuzumab to treat BC in September 1998, following a clinical trial in HER2+ metastatic patients in which the addition of trastuzumab to chemotherapy significantly improved the median progression-free survival (PFS) to 7.4 from 4.6 months and the median overall survival (OS) to 25.1 from 20.3 months, as compared with chemotherapy alone [3].

Despite the success of trastuzumab in improving outcomes, a high fraction of the patients demonstrate primary resistance and those who respond initially develop resistance to it within a year [4]. Furthermore, patients can exhibit multiple types of resistance, which can be clinically difficult to identify, particularly in clinical settings in which trastuzumab in used in combination with other agents. In this article, we briefly review the mechanisms of action engaged by trastuzumab to mediate its anti-tumor effects (Figure 1) and then focus in more detail on the molecular and cellular mechanisms of trastuzumab resistance, including the induction of immune suppression, vascular mimicry, the generation of breast cancer stem cells, the deregulation of long non-coding RNAs, and metabolic escape. We discuss how these resistance mechanisms can lead to the development of customized treatment combinations that could augment trastuzumab-based therapies and result in improved therapeutic outcomes.

## 2. Cellular and Molecular Mechanisms of Action of Trastuzumab

### 2.1. Trastuzumab Alters HER2 Receptor Dimerization and Cell Surface Expression

The oncogenic activity of HER2 is mediated by HER2 homodimers or HER2/HER1, HER2/HER3, or HER2/HER4 heterodimers [5]. HER2 is also known to form heterodimers with other kinases, such as insulin-like growth factor 1 receptor (IGF-1R), particularly in trastuzumab-resistant tumor cells, as discussed below [6]. Whereas heterodimers activity can be modulated by ligand, homodimers demonstrate constitutive activation. Despite its extensive use and study, how trastuzumab modifies signaling through HER2 remains unclear. It is accepted that trastuzumab has little effect on ligand-induced HER2 heterodimerization [7,8,9]. However, controversy remains as to whether trastuzumab disrupts ligand-independent homo- or heterodimerization. While Junttila et al. and Gaborit et al. reported disruption of heterodimers, Diermeier et al. found no evidence of this in HER2-overexpressing cell lines [8,10,11]. One possible solution to the observed discrepancies in studies could come from computer modeling, which has suggested that trastuzumab leads to subtle changes in the flexibility of the kinase domain, imparting both antagonistic and agonistic activity, but without complete complex disruption [12].

The clinical efficacy of trastuzumab has also been linked to the blockade of cleavage of the HER2 extracellular domain (ECD). HER2+ BCs are known to shed the ECD through proteolysis and production from an alternative transcript [13,14,15]. HER2 cleaving metalloproteinases, such as ADAM10 and matrix metalloproteinases (MMPs), are upregulated in the BC tumor microenvironment, which can result in the generation of the NH2-terminal truncated HER2 protein, termed p95HER2 [16,17,18,19,20,21,22,23,24]. P95HER2 possesses constitutively active kinase activity, is observed in 20–50% of HER2+ tumors, and elevated expression is associated with poorer outcomes [19,25]. Molina et al. demonstrated that trastuzumab, but not pertuzumab, which recognizes a distinct epitope, was able to block cleavage of full-length HER2, likely due to steric hindrance of the cleavage site [19]. Some clinical studies have reported a reduced extracellular domain of HER2 in serum samples from patients who received trastuzumab for their treatment [26,27]. Subsequent studies by Liu et al. showed further support for the cleavage hypothesis in that the co-blockade of ADAM kinases during treatment of HER2+ BC in vivo and in vitro augmented the activity of trastuzumab [16]. The effectiveness of such a combination may, however, be limited, as other studies have observed that soluble ECD itself acts as an important decoy by binding to full-length HER2 and preventing HER2 homo- and heterodimer formation and tumor cell proliferation [28]. Overall, the observations support the idea that the effectiveness of trastuzumab could be due in part to its blockade of HER2 ectodomain cleavage, which reduces production of oncogenic p95HER2 [29].

One early hypothesis, driven by early murine anti-HER2 studies in mice, was that the efficacy of trastuzumab suppressed the growth of tumors by reducing cell surface HER2 levels through endocytosis [30,31,32]. HER2, unlike its other family members, however, resists internalization and degradation [33]. Stability is maintained by the complexing of HER2 with Ezrin/NHERF1/PCMA2 in MAL2-containing lipid rafts, notably in internalization resistance membrane protrusions [34,35]. Although HER2 is internalized, alone or in complex with trastuzumab, it is slow and most receptor is rapidly recycled back to the cell surface unless it is extensively crosslinked [35,36]. It continues to be a matter of debate as to whether trastuzumab has any impact on internalization or intracellular trafficking through a recycling pathway or a degradative pathway (reviewed in [7]). While many studies show that trastuzumab (or its parental murine, non-humanized antibody, 4D5) does reduce expression of HER2 in human cell lines, other studies have refuted the cell surface downregulation hypothesis [29,37,38,39,40,41,42]. One potential explanation is that differential HER2 levels among various cell lines could impact results. In this regard, Ram et al. found that internalization and degradation were dependent on the density of HER2 at the cell surface [43]. Whereas high-level expression was associated with rapid recycling, increased degradation and cellular depletion of HER2 were observed in cells with lower cell surface receptor expressions. Shi et al. found that engagement of immune effectors was required for downregulating the receptor, both in vitro and in vivo in mice, by supplying IFN-γ, which subsequently led to STAT1 activation and subsequent modulation of intracellular HER2 trafficking. Clinical studies also support that at least a subset of HER2 BCs are susceptible to HER2 degradation and loss. Ignatov et al., for example, recently observed that neoadjuvant trastuzumab treatment was associated both with loss of HER2 expression in ~47% of cases and poorer disease-free survival [44]. Although it could be argued that tumor cell heterogeneity resulted in the discordance, the additional observation that the interval between treatment and determination of HER2 expression in the second histology was associated with concordance (i.e., a longer interval was associated with better concordance) supports transient loss due to internalization and degradation. Understanding the molecular interactions of trastuzumab with HER2 and the resulting cellular response continues to be an important topic and has potential to inform clinical decisions and the further development of trastuzumab-based therapeutics such as the antibody drug conjugate trastuzumab emtansine (TDM-1) [45]. 

### 2.2. Trastuzumab Attenuates/Modifies Proximal HER2 Signaling

Trastuzumab (or its parent murine monoclonal antibody) has been shown to suppress HER2 signaling in HER2-amplified breast cancer cells by blocking the interactions of Src kinase with HER2, thus reducing Src-Y416 phosphorylation and activation [46]. Reduced Src activity correlates with reduced tyrosine phosphorylation and increased membrane binding of the phosphatase PTEN, which results in rapid Akt dephosphorylation. This mechanism of trastuzumab is supported by several, but not all, clinical studies showing that PTEN negative tumors are unresponsive to trastuzumab, including breast and gastroesophageal cancers [46,47,48,49,50,51]. Although still debatable, Akt activation may be further suppressed by trastuzumab by preventing HER2-mediated phosphorylation of HER3, thus eliminating phosphatidylinositol 3-kinase (PI3K) p85 subunit (p85) binding, p85/p110 (p110 is the PI3K catalytic subunit) assembly, and subsequent Akt phosphorylation by PDK1 and mTORC2 [7]. In addition to the Akt pathway, other signaling pathways are attenuated by trastuzumab in HER2-overexpressing tumor cells, which may contribute to the therapeutic effectiveness of the antibody. HER2 homodimer-mediated MAPK pathway activation is blocked by trastuzumab through a dissociation of the SHC adaptor from MAPK [52]. Attenuated MAPK is associated with increased tumor apoptosis and reduced sensitivity to mitogenic signals (e.g., TGFβ) [53]. While several studies have suggested that the efficacy of trastuzumab is due in part to signaling blockade, other studies have refuted this hypothesis, suggesting that signaling is largely unaffected by trastuzumab due to extensive overexpression and the uncoupling of extracellular domain regulation of intracellular kinase activity [54]. The debate is complicated by observations that trastuzumab does not block, but stimulates HER2 phosphorylation, indicative of activation rather than inhibition [55,56,57]. Despite this uncertainty, many publications report that trastuzumab directly blocks proliferation of HER2+ cancer cells (see below), thus indicating signal disruption [7,8,52,55,57,58,59].

### 2.3. Trastuzumab Alters the Cellular Physiology of HER2-Overexpressing Tumor Cells

A key facet of trastuzumab is its ability to block proliferation by inducing cell cycle arrest in the G1 phase. Lane and colleagues reported in their murine model that an anti-HER2 antibody interfered in the HER2 receptor signaling, which subsequently deregulated p27 sequestration by downregulating proteins such as D-type cyclins and c-Myc that are involved in the p27^Kip1^ sequestration process [60]. The released p27 bound to and inhibited the cyclin E/Cdk2 complexes that drive cells through the G1/S phase, and reduced proliferation. Le et al. corroborated that trastuzumab treatment can induce p27^Kip1^ upregulation and G_1_ cell cycle arrest resulting in the growth inhibition of breast cancer cells [61]. Additionally, they showed that the G_1_ cell cycle arrest was dependent on both the time and magnitude of the p27^Kip1^ protein induced. Consistent with this, trastuzumab, when used in the adjuvant setting, appears to be less effective in individuals with elevated pretreatment levels of p27kip1 [62]. These studies indicate that the expression of p27 could potentially be a biomarker for trastuzumab efficacy. Trastuzumab also induces increased expression of miR-26a and miR-30b, which in turn downregulates cyclin E2 [63]. Papadakis reported that increased expression of the co-chaperoneBcl-2-associated athanogene 1 (BAG-1) protein correlated with HER2 expression and was essential for the optimal growth of some HER2-overexpressing breast cancer cells. The combination treatment of trastuzumab and small molecule inhibitors targeting BAG-1 attenuated protein synthesis, suppressed ERK and AKT signaling pathways, and induced G1/S cell cycle arrest [64].

HER2 overexpression is correlated with increased expression of several anti-apoptotic proteins including Bcl-2 family members and MDM2 [65,66]. Trastuzumab has been shown to down modulate these anti-apoptotic proteins [53]. Trastuzumab also blocks key DNA repair systems, such as p21/WAF1, which are critical to tumor cell survival after exposure to DNA damage induced by cytotoxic drugs cisplatin and doxorubicin or radiation [67,68,69,70,71]. Co-targeting both HER2 and ER leads to the induction of apoptosis in breast cancer cells with both HER2+ normal and overexpression [72]. 

### 2.4. Trastuzumab Limits Formation of Aggressive Tumor Microenvironments

Overexpression of HER2 induces neuregulin 1 (NRG1)- and hypoxia-inducible factor 1 (HIF-1)-dependent vascular endothelial growth factor (VEGF) release [73,74,75]. HER2+ breast cancers stimulate VEGF-mediated vessel production for improved nutrient and oxygen delivery. The high levels of VEGF release, however, results in a chaotic vascular network that results in selection of more invasive and aggressive cells that are resistant to cytotoxic chemotherapy [76]. Trastuzumab suppresses the release of VEGF and significantly improves perfusion [77,78]. Trastuzumab may also block release of other mediators of angiogenesis, including Ang-1, PAI-1, and TGF-α [78].

### 2.5. Trastuzumab Activates Innate and Adaptive Immune Responses

In addition to its direct effects on intrinsic tumor cell biology, trastuzumab has been shown to engage both innate and adaptive immune responses. As a full-length humanized IgG1 antibody, trastuzumab contains an Fcγ region, enabling binding to various Fcγ receptors (FcγRs). There are multiple FcγRs, including both activating and inhibitory receptors, whose expression is confined to immune cells in humans, including B cells, DCs, NK cells, macrophages, and neutrophils. FcγR expression across a broad spectrum of critical immune effectors suggests the potential for multiple immunologic mechanisms in the clinical action of trastuzumab. From a clinical perspective, the role of FcγRs is relevant to individual patient response to trastuzumab, as indicated by findings that common polymorphisms in *FCGR* genes have been shown functionally to alter the binding affinity of trastuzumab [79,80,81]. Some genetic association studies of *FCGR* polymorphisms have shown improved survival in patients with high-affinity alleles (i.e., alleles predicted to have increased binding affinity to trastuzumab), but other studies have failed to replicate these findings [79,82,83,84,85]. 

Antibody-dependent cellular cytotoxicity (ADCC) is a key mechanism of action of trastuzumab. ADCC occurs when an antibody is bound to the tumor cell and the Fcγ region of the tumor-bound antibody is recognized by immune cells expressing FcγRs [86]. This results in a directed release of lytic granules or an induction of death-receptor-mediated apoptosis via the expression of Fas ligand or TRAIL [87]. Clynes et al. first demonstrated a role of FcγRs in ADCC by showing that trastuzumab was ineffective in mice that lacked the activating FcγRIIIa gene and that the antibody was more effective in mice that lacked the inhibitory FcγRIIb [86]. ADCC is mediated by multiple cell types including neutrophils, macrophages, eosinophils, and natural killer (NK) cells [88,89]. While the relative contribution of each of these mediators of ADCC in the human setting is unclear, it has been observed that trastuzumab-containing regimens induce infiltration of both NK cells and macrophages into tumors [90]. Furthermore, patients with pathologic complete responses to neoadjuvant paclitaxel and trastuzumab therapy demonstrate increased activation of NK cells, as assessed in the peripheral blood [91]. It is also notable that while neutrophils may be able to mediate ADCC, the unique and high-level expression of the human decoy FcγRIIIb receptor in some individuals (due to gene duplication) may limit their contribution to the clinical activity of trastuzumab [92].

Antibody-dependent cellular phagocytosis (ADCP) by macrophages has recently been proposed as a potential mechanism of trastuzumab action. Cancers are often found to harbor large numbers of tumor-associated macrophages, which, despite their propensity to drive tumor progression, still retain FcγR-mediated phagocytic capacity [93]. Preclinical studies in other monoclonal antibody therapeutic settings show that ADCP is essential for therapeutic efficacy [94,95,96]. In murine and in vitro human cell culture models, trastuzumab activates phagocytic killing through murine FcγRIV and human FCγRIII, respectively [97]. Furthermore, in murine models, ADCP and not ADCC may be the dominant innate effector mechanism responsible for tumor blockade [98]. In addition to ADCP, macrophages (and other immune cells) also kill opsonized targets through a process called trogocytosis, a form of biting or nibbling [99]. Velmurugan et al. showed that persistent trogocytic attack results in the killing of trastuzumab-coated HER2-overexpressing breast cancer cells [100]. The role of ADCP or trogocytosis in the human clinical setting is unclear, but has some impact, given the known infiltration of HER2+ breast cancer by macrophages, which is augmented upon trastuzumab administration [90,101]. Furthermore, trogocytosis-mediated expression of HER2 on immune cells in tumor tissue is associated with pathologic complete response in patients treated with trastuzumab [102].

Trastuzumab is an IgG1 subclass antibody and as such can activate the classical complement pathway, leading to local generation of inflammatory mediators, complement-mediated cell phagocytosis, and complement-dependent cytotoxicity (CDC) mediated by the membrane attack complex (MAC) [103]. While previous studies have stated that CDC, while induced, is not a major mechanism of action of trastuzumab, recent studies indicate that CDC is important in the activity of combined trastuzumab and pertuzumab therapy. Tsao et al. found that the therapeutic efficacy of combined trastuzumab and pertuzumab, murine models, involved both CDC and CDCP of HER2+ tumors [104]. The team also demonstrated that tumor C1q expression was positively associated with survival outcome in a cohort of HER2+ breast cancer.

Trastuzumab is also known to stimulate adaptive immunity including generation of both T and B cells. Our group first demonstrated this in patients treated in both the metastatic and adjuvant settings, where we observed that treatment with trastuzumab in combination with chemotherapy significantly increased endogenous circulating anti-HER2 antibodies and helper T cells [105]. Of the 22 metastatic patients in that study, those that demonstrated objective clinical responses exhibited more frequent and larger anti-HER-2/neu humoral responses. In other cohorts of patients (both metastatic and adjuvant), we subsequently demonstrated that the generation of HER2-specific antibody immunity was associated with increased recurrence-free, progression-free, and overall survival [106,107]. Further studies demonstrated that the adaptive immune responses were dependent on the presence of trastuzumab since comparator patients treated with chemotherapy alone did not show evidence of induction of anti-HER2 immunity. We recently reported similar results in patients that were treated with margetuximab, a trastuzumab derivative with enhanced FcγR-binding activity [108]. This vaccinal effect of trastuzumab is likely mediated either by antigen-presenting cell cross-presentation of trastuzumab-bound HER2, which has been released into the tumor microenvironment and/or by trastuzumab-induced endocytosis and MHC class I presentation directly on tumor cells [109]. Studies have shown that trastuzumab is able to activate CD8 T cells by increasing the levels of MHC class I bound HER2 peptide at the cell surface through augmented endocytosis and proteasomal degradation [37,110]. While a definitive role of adaptive immunity has been clearly established in murine using the parental 4D5 antibody, whether the adaptive immune response has a major role in the clinical efficacy in humans remains debatable at present [111].

## 3. Molecular and Cellular Mechanisms of Resistance to Trastuzumab

Despite several mechanisms of action and clear clinical benefit, a fraction of adjuvant patients and most metastatic patients demonstrate disease progression due to primary or adaptive resistance. For example, a retrospective study of patients with locally advanced or metastatic HER2+ breast cancer who were treated with trastuzumab in the first-line setting between 2001 and 2010 indicates objective response rates (ORRs) of only 65% [112]. This and similar studies underscore the importance of understanding the mechanisms of resistance, of which several are discussed below, and identifying approaches to improve outcomes. Figure 2 is a schematic representation of the different mechanisms of resistance to trastuzumab.

### 3.1. Vascular Mimicry and Hypoxia

Altered angiogenesis has been demonstrated to be involved in acquired resistance to trastuzumab. Vasculogenic mimicry (VM) is a recently identified strategy where tumor cells themselves transition from an epithelial phenotype to a more vascular phenotype characterized by the presence of endothelial cell-surface antigens such as periodic acid-Schiff (PAS), but in the absence of classical markers such as CD31 and tube formation [113]. VM has been documented in breast cancer and is associated with a more malignant phenotype in triple-negative and HER2-positive tumors [114]. Surgically resected tumor tissues from HER2+ BC patients treated with trastuzumab along with neoadjuvant chemotherapy (NAC) showed a significantly higher count of VM channels per unit area as compared to the respective controls [113]. Similarly, in paired tumor samples collected before and after neoadjuvant treatment, a significant increase in the VM channels was observed in trastuzumab-resistant tumor cells derived from patients that received trastuzumab in addition to chemotherapy, as compared to chemotherapy-naïve specimens. In vitro mechanistic investigations showed that CD144 blockade did not restore trastuzumab sensitivity. Rather, resistance was due to the activation of alternate oncogenic receptors and pathways including EGFR, IGF-1R, VEGF2, and FGFR2 due to prolonged HER2 blockade. Moreover, salinomycin efficiently suppressed VM by compromising actin cytoskeleton integrity inhibition of Rho-GTPases [113]. Of note, Salinomycin has also been shown to target cancer stem cells [115], suggesting their involvement in VM and trastuzumab resistance.

### 3.2. The Role of Breast Cancer Stem Cells (BCSCs) in Trastuzumab Resistance

Several studies have characterized trastuzumab-resistant breast cancer stem cells (BCSCs) as drivers of metastasis and recurrence. It has been previously shown that breast cancer cells expressing CD24^−/low^/CD44^+^ are termed cancer-initiating cells or BCSCs [116,117]. A prospective study isolated BCSCs (identified as CD44^+^CD24^−/low^lineage^−^ cells) from eight out of nine patients and demonstrated their ability to form tumors in immunodeficient patients with only 100 cells [116]. The resistance of these cells to radiation therapy and chemotherapy has well-documented in both preclinical and clinical models [117,118,119]. The self-renewal and re-differentiation capability of BCSCs are essential for tumor growth and progression. They acquire these properties through the modulation of several fundamental processes in their favor such as increased metabolism, induction of resistance to apoptosis and autophagy, stimulation of stem cell transcription factors, upregulation of proteins involved in drug transport, and deregulation of detoxification mechanisms. They achieve this through the deregulation of multiple signaling pathways as well as epithelial to mesenchymal transition [117,120]. 

Although Magnifico et al. previously demonstrated that trastuzumab effectively targeted BCSCs due to their increased HER2 expression [121], BCSCs have conversely been implicated in innate and acquired resistance to trastuzumab. In vitro, chronic treatment of BT474 spheroids with trastuzumab increased the BCSC population, contributing to the development of trastuzumab resistance. BT474 spheroids had higher expression of HER2 compared to 2D cell cultures. There were two subpopulations of cells expressing different levels of HER2 in these spheroids. The population with higher HER2 expression increased following trastuzumab treatment and correlated with trastuzumab resistance. BCSCs in these spheroids also had higher HER2 levels post-trastuzumab treatment compared with the controls. A limitation of this study was that these results were not observed in MCF7 spheroids, their second model. The authors postulated that this could be because MCF7 cells lack sensitivity to trastuzumab. In the MCF7 spheroids, the expression of HER2 in most of the cells was low and only ~28% of the BCSCs expressed cell surface HER2 [122]. 

Several specific molecules and pathways have recently been linked to both stemness and trastuzumab resistance. One example is the receptor tyrosine kinase EPHA5. Li et al. found that expression of EPHA5 is downregulated in trastuzumab-resistant patients and correlates with a poor prognosis. Mechanistically, they demonstrated that reduced expression of EPHA5, by knockout, was correlated with increases in BCSC-like properties, as characterized by increased cells with the CD44+/CD24low phenotype and mammosphere-forming ability, as well as increased expression of CD133+ and NANOG and decreased expression of E-cadherin. Further, the activation of Notch-1 and PTEN/AKT signaling pathways was observed. The study also showed that downregulation of EPHA5 in HER2+ breast cancer cells induced trastuzumab resistance, whereas forced overexpression led to increased sensitivity. These findings indicate that in HER2+ BC, EPHA5 could be evaluated as a predictor of response to trastuzumab therapy and, additionally, combining it with a Notch1 pathway inhibitor may augment anti-tumor responses [123]. This finding is supported by other studies that showed that Notch signaling supports breast cancer growth by promoting the BCSC phenotype, fostering resistance to therapy [118,124,125]. To support the concept that Notch blockade in combination with trastuzumab could improve responses, Osipo demonstrated that combination therapy with trastuzumab and a Notch-1 small-interfering RNA (siRNA) inhibited the growth of both trastuzumab-sensitive and -resistant cells [126]. Additionally, the investigators also found that Notch inhibition using gamma-secretase inhibitor (GSI) sensitized resistant cell-line trastuzumab. 

Cyclin-dependent kinase 12 (CDK12) is traditionally known for its role in DNA repair [127]. In breast cancer, it is sometimes concurrently amplified along with HER2 [128]. High CDK12 and HER2 expression correlates with trastuzumab resistance, recurrence of the disease, and poor survival [128]. Mechanistically, CDK12 promotes the self-renewal of CSCs and trastuzumab resistance through the activation of Wnt signaling pathways, rather than through its role in DNA damage. Dinaciclib-mediated inhibition of CDK12 kinase activity resulted in anti-tumor effects in trastuzumab-resistant HER2+ cancer [129]. Consistent with this, Wnt signaling has been found to be higher in patient-derived metastatic cancer stem-like cells than non-stem tumor cells [130]. Activated Wnt signaling results in the transactivation of EGFR, subsequently leading to BCSC-inducing EMT, thereby contributing to trastuzumab resistance [131]. Another therapeutic option for increasing the sensitivity of HER2+ breast cancer by blocking BCSCs and Wnt signaling is the use of Geldanamycin, a heat shock protein-90 kDa (HSP-90) inhibitor [132]. 

In HER2+ BC, HER2 overexpression is known to mediate IL-6 secretion along with inflammatory transcriptional signatures. IL-6 activates an autocrine JAK1-STAT3 signaling loop that in turn increases STAT3 activation in HER2+ breast cancer cells [133]. In pancreatic cancer, HER2 also stimulates the JAK pathway through the activation of Src kinase and Src-mediated STAT2 nuclear translocation in the pancreatic cancer model [134,135]. Marotta et al. demonstrated that IL-6/JAK2/STAT3 signaling was particularly upregulated in CD44^+^CD24^−^ BCSCs relative to other tumor cells and that JAK2 inhibition significantly reduced this population as well as tumorigenicity [136]. Another study has also shown that STAT3 signaling enhances the mammosphere-forming efficiency and tumor-initiating capability of BCSCs [137]. Thus, data from several studies indicate that HER2 can positively upregulate STAT3 signaling, leading to induction of BCSCs, thereby potentially contributing to trastuzumab resistance. 

Telomerase has also been implicated in the development and maintenance of BCSCs and other cancer stem cells [138]. Goldblatt et al. initially showed that the telomerase template antagonists, imetelstat, can restore sensitivity to trastuzumab [139]. Similarly, Koziel et al. observed similar outcomes in subsequent studies and demonstrated a reduction in the stem cell population and their functional ability to form mammospheres and generate metastatic lesions [140]. 

Trastuzumab resistance may also depend on non-tumor cells in the tumor microenvironment. Breast and other cancers consist of various cell types such as endothelial cells, fibroblasts, immune cells, and mesenchymal stem cells (MSCs). MSCs are multipotent stromal cells that are known to accumulate in tumors because of local inflammatory responses [141]. MSCs have been demonstrated to induce trastuzumab resistance through lncRNA AGAP2-AS1-mediated stemness. In that study, an analysis of serum samples from 90 trastuzumab-treated patients (45 responding and 45 non-responding) indicated that exosomal AGAP2-AS1 is overexpressed in patients who are resistant to trastuzumab compared to patients who respond. Mechanistic investigations revealed that MSCs can induce AGAP2-AS1, which subsequently increases the expression of carnitine palmitoyltransferase 1 (CPT1) by binding with miR-15a-5p and forming a complex with human antigen R. This complex in turn stabilizes CPT1 mRNA. The increased CPT1 enhances fatty acid oxidation and subsequently increases the expression of BSCS-related genes and characteristics [142].

Lastly, in addition to trastuzumab resistance in BCSCs, studies on other cancers have provided insights into the role of stemness. For example, In HER2+ gastric cancer cells, acquired trastuzumab resistance has been shown to be mediated by GSE1 through the induction of stem-cell-like characteristics. Trastuzumab-resistant cells had higher GSE1 expression and the sensitivity to trastuzumab was restored upon GSE1 knockdown. GSE1 upregulated the expression of BCL-2 and enhanced anchorage-independent spheroid formation, hallmarks of stemness [143]. The expression of human Gse1 coiled-coil protein, GSE1, in HER2+ gastric cancer is associated with poor pathologic features typically associated with poorer survival, including lymph node metastasis and advanced disease stages. 

Collectively, these studies indicate that one mechanism of trastuzumab resistance is through the induction of BCSCs and possibly CSCs in other cancers. Conventional therapies such as chemotherapy, endocrine therapy, and radiotherapy also have limited activity against BCSCs, thus suggesting that the induction of trastuzumab resistance can lead to broader resistance to subsequent therapies. Thus, an increased understanding of how trastuzumab induces the BCSC phenotype is an important objective, which is the subject of recent reviews [144,145]. 

### 3.3. Trastuzumab Resistance Is Associated with Metabolic Changes

Evidence over the last decade also suggests that deregulation of metabolism contributes significantly to trastuzumab resistance. Zhao et al. initially demonstrated that HER2 overexpression leads to increased synthesis of heat shock factor 1 (HSF1) and formation of the HSF-1 trimer, which activates synthesis of lactate dehydrogenase A (LDH-A), a critical enzyme in converting pyruvate to lactate during anaerobic glycolysis [146]. In subsequent studies, the same group found that trastuzumab treatment of HER2-expressing tumor cells significantly reduced HSF1 and LDH-A expression, reducing glycolysis and suppressing cell growth [147]. In contrast, forced overexpression of HSF1 and LDH-A led to trastuzumab resistance. t-Darpp is known to facilitate trastuzumab resistance through promoting HER2 protein stability and activation of other signaling pathways (discussed in detail below) [148]. Lenz et al. showed that t-Darpp is also involved in glycolysis-mediated resistance to trastuzumab [149]. Their studies suggest that t-Darpp, which is upregulated in trastuzumab-resistant cells, activates IGF-1R signaling through heterodimerization with HER2 or EGFR, stimulating glycolysis and conferring resistance. Alternatively, recent work by Gale et al. suggests that resistance may be mediated by the increased reliance of trastuzumab-resistant tumor cells on oxidative phosphorylation. In that study, transcriptomic profiling revealed that resistant cells exhibited increased expression of ATP synthase genes and demonstrated dependency on ATP synthase function. Importantly, they found that combining the ATP synthase inhibitor oligomycin A with trastuzumab led to the regression of trastuzumab-resistant tumors in vivo [150]. Thus, a novel therapeutic strategy to enhance efficacy in both trastuzumab-sensitive and -resistant patients may be to combine trastuzumab with tumor-targeted glycolysis or aerobic respiration inhibitors (e.g., 2-DG or oxamate), strategies that are currently under development [147,151,152].

Fatty acid metabolism is also central to HER2 oncogenicity in many ways. Fatty acid synthase (FASN) is a critical enzyme that catalyzes the intracellular synthesis of palmitate from acetyl-CoA and malony-CoA. Menendez, et al. reported that FASN activity was essential for the expression and oncogenic activity of HER2 [153]. That study reported that silencing of FASN upregulates PEA3, a transcriptional repressor of HER2, resulting in significant loss of surface HER2 expression. Conversely, HER2 also upregulates expression of FASN, further empowering endogenous generation of fatty acids [154]. FASN blockade also significantly reduces shedding of the HER2 extracellular domain, thus possibly reducing unrestrained oncogenic activity of HER2p95 [155]. More recent studies have demonstrated that fatty acid metabolism is essential for the brain metastasis of HER2+ breast cancer, possibly mediated by high levels of fatty acid binding protein 7 (FABP7), which supports formation of lipid storage [156,157]. Trastuzumab treatment is known to suppress FASN expression, but this may be compensated by a metabolic switch from endogenous lipogenesis to external fatty acid uptake during the development of the resistant phenotype [154]. This potential metabolic switch was recently exemplified by Feng et al., who showed that acquired resistance to the HER2 inhibitor lapatinib is mediated by upregulation of CD36, a transmembrane glycoprotein that acts as a transporter for, among other ligands, long-chain fatty acids [158]. Although that study did not conclusively demonstrate that trastuzumab resistance was also accompanied by upregulation of CD39, a subsequent report by Ligorio showed that high baseline expression of CD39 was associated with poorer event-free survival in patients treated with a combination of paclitaxel and trastuzumab, suggesting an increased uptake of long-chain fatty acids contributes to primary resistance to trastuzumab [159].

Another way that lipid metabolism is involved in HER2 pathogenesis is through lipid rafts, which are membrane microdomains involved in various cellular processes, such as signaling and trafficking. These rafts are particularly important in amplifying HER2 oncogenic signaling and are formed in part through MAL2, a lipid raft resident protein, which is highly upregulated in HER2+ BC cells [33]. Through its role in the lipid raft formation, MAL2 is critical for membrane retention and signaling mediated by HER2. MAL2 appears to be essential for the formation of HER2–EGFR heterodimer complexes and their retention at the cell surface. MAL2 expression has been also shown to be integral for the structural and functional association between Ezrin, HER2, phosphatidylinositol 4,5-bisphosphate (PIP2), and PI3K/pAKT within the membrane that in turn is essential for the oncogenic signaling. MAL2 lipid rafts are also needed for PMCA2/HER2 interactions, which prevent high intracellular calcium concentrations that can disrupt HER2-mediated signaling. Furthermore, trastuzumab-resistant cell lines have increased HER2/MAL2 interactions in the lipid rafts. The importance of lipid metabolism in HER2+ breast cancer and resistance to trastuzumab therapy suggests that lipid metabolism inhibitors might improve therapeutic outcomes. Indeed, in vitro studies have shown that the combination of trastuzumab and FASN inhibitor cerulenin leads to synergistic induction of apoptosis [153].

### 3.4. Activation of Alternative Signaling Pathways

Several clinical studies have reported correlations between the receptor-mediated deregulation of various signaling pathways and trastuzumab resistance, particularly PI3K/Akt/mTOR and MAPK pathways. Molecular mechanisms attributed to this aberrant signaling include signaling through other HER receptors, upregulation of other receptors, and activation of alternate signaling pathways. In addition to HER2, there are other receptors in the HER family, such as EGFR/HER1, HER3, and HER4, that have the potential to form dimers and mediate signaling; therefore, trastuzumab cannot completely mitigate the signaling [160]. Results from a study conducted in 155 patients treated with trastuzumab after the development of metastasis or as adjuvant/neoadjuvant treatment showed a correlation between trastuzumab resistance and either deregulation of the PI3K signaling pathway and/or EGFR and IGF-1R overexpression in about 25% of the tumor samples from patients with HER2+ breast cancer [161]. Another retrospective study conducted on tumor sections from 807 breast cancer patients provided further clinical evidence for the association between HER2 activation and aberrant EGFR expression. That study showed EGFR overexpression in 35% of the 306 HER2+ breast cancer specimens [162]. Notably, they also observed that 97% of tumors with activated HER2, as assessed by HER2 phosphorylation, co-overexpressed EGFR. Another study in patients with high-risk primary breast cancer demonstrated that EGFR expression was associated with worse prognosis, specifically when co-expressed with HER2 [163]. Various studies have reported that elevated expression of other HER family ligands leads to resistance to trastuzumab monotherapy. For example, co-expression of EGF in HER2 cell lines alleviates the growth inhibitory effect of trastuzumab by modulating EGFR/HER2 homo- and hetero-interactions [8]. Another preclinical study reported that PKI166, a bispecific HER1/HER2 kinase inhibitor, more effectively inhibited EGF-associated ligand-induced bypass of the trastuzumab-mediated proliferation blockade, as compared with other monospecific agents [164]. 

The PTEN/PI3K/Akt/mTOR and MAPK pathways can also be activated by other non-HER receptor tyrosine kinases. For example, one that is important in resistance is the insulin-like growth factor I receptor (IGF-1R), which is overexpressed in several solid tumors including in about 43–50% of primary breast cancers [165]. IGF-1R expression is associated with worse prognosis in breast cancer [166]. Depending on subtype, IGF-1R activates both the MAPK and pERK1/2 and the PTEN/PI3K/Akt/mTOR pathways [165]. IGF-1R has been shown to mediate trastuzumab resistance through the direct activation of HER2. For example, in trastuzumab-resistant breast cancer cells, IGF-1R and HER-2 form a heterodimer, which can ligate IGF-1 and lead to HER2 phosphorylation. IGF-1R blockade disturbed the IGF-1R/HER2 interactions and restored trastuzumab sensitivity [167]. IGF-1R signaling is also reported to mediate trastuzumab-resistant HER2+ breast cancer cells through cell cycle modulation. IGF-1R signaling has been shown to enhance ubiquitin-mediated p27^Kip1^ degradation, which results in increased Cdk2 activity and subsequent release of cells from G_1_ arrest [168]. 

The MET receptor and its ligand, hepatocyte growth factor, contribute to trastuzumab resistance by inhibiting trastuzumab-mediated p27 induction and continued AKT activation [169]. Receptor tyrosine kinase Eph receptor A2 (EphA2) is another example [170]. The increased expression of EphA2 has been reported to contribute to both intrinsic and acquired trastuzumab resistance. Trastuzumab treatment in the resistant cells activates EphA2, which leads to the stimulation of Src kinase and amplified PI3K/Akt and MAPK signaling. Additionally, increased EphA2 expression in HER2+ breast cancers is correlated with poor prognosis. Another receptor implicated in trastuzumab resistance is the receptor for erythropoietin (EpoR). In preclinical models, it was demonstrated that recombinant human erythropoietin (rHuEPO) protected breast cancer cells against the anti-tumor effects of trastuzumab treatment through the activation of Src facilitated by JAK2 and PTEN inactivation. Retrospective analysis conducted by the same group supported this finding that concurrent administration of rHuEPO conferred trastuzumab resistance in HER2+ breast cancer patients [171].

The catecholamine/β2-adrenergic receptor (β2-AR) signaling pathway has been shown to facilitate trastuzumab resistance [160]. In a gastric cancer model, catecholamine-mediated activation of β2-adrenergic receptor (β2-AR) resulted in the upregulation of MUC-4 expression through the activation of STAT3 and ERK. MUC-4 impaired the recognition and physical binding of trastuzumab to the HER2 receptors, thereby contributing to trastuzumab resistance [172]. Liu et al. further demonstrated that HER2 overexpression and increased phosphorylation of ERK resulted in the release of epinephrine, which in turn elevated β2-AR expression in an autocrine manner [173]. The catecholamine enhanced HER2 mRNA expression and isoproterenol stimulated the activation and nuclear translocation of STAT3. This resulted an increase in the expression of miR-21 and MUC-1, which led to PTEN insufficiency and subsequent activation of PI3K and Akt. Furthermore, mammalian target of rapamycin (mTOR) was activated through inhibition of miR-199a/b-3p. Taken together, these results show that β2-AR signaling influences pathways involved in trastuzumab resistance.

Primary and acquired resistance to trastuzumab may also involve the estrogen receptor (ER). Wang et al. indicated that the ER pathway can act as a survival pathway in cells positive for both estrogen receptor and HER2 [174]. It has been observed in some neoadjuvant clinical trials that ER+ HER2+ tumors that were treated with a combination of chemotherapy and anti-HER2 drugs had lower pathological complete response compared to ER tumors [175]. The phase III clinical trial, TAnDEM, showed that combination treatment of trastuzumab and anastrozole improved the outcomes for metastatic HER2/hormone receptor dual-positive breast cancer patients as compared to anastrozole treatment alone [176]. A recent phase II clinical trial, NA-PHER2, explored the potential of triple targeting of ER, HER2, and RB1 in HER2+ and ER+ breast cancer in the neoadjuvant setting [177]. The findings of this study suggest that combination treatment of pertuzumab, palbociclib, trastuzumab, and fulvestrant shows promise as an effective chemotherapy-free treatment. A total of 29 of the 30 patients achieved objective clinical responses before surgery and 8 of those patients showed a pathologic complete response in the breast and axillary nodes. 

### 3.5. HER2 Molecular Variants Contribute to Therapeutic Resistance

Several molecular variations in the HER2 protein impede its ability to bind trastuzumab and thereby contribute to trastuzumab resistance. Improved knowledge of these variants has directed treatment regimens towards combining trastuzumab regimens with additional anti-HER2 therapies that target these variants. For example, p95HER2, also known as the HER2 carboxy-terminal fragment, lacks the extracellular domain that is essential for trastuzumab binding. As discussed above, this fragment is generated through either extracellular domain shedding by the metalloprotease ADAM10 or through the alternative initiation or translation of the HER2-coding mRNA. p95HER2 promotes trastuzumab through its capability to constitutively form homodimers, which are stabilized by disulfide bonds [178]. In metastatic breast cancer patients treated with trastuzumab, high levels of p95HER2 were found to correlate with shorter progression-free survival and overall survival [179]. Thus, p95HER2 is a valid target for a subset of HER2+ patients with metastatic breast cancer. 

A splice variant of HER2, HER2Δ16, that lacks exon 16, which encodes a portion of the extracellular region, is widely expressed and represents about 2–9% of the total HER2. It is also thought to contribute to trastuzumab resistance. HER2Δ16 is ten times more potent in inducing cellular transformation than the full-length HER2 and less sensitive to trastuzumab, potentially because the homodimers formed by these variants are strengthened by disulfide bonds and this weakens the reactivity of trastuzumab [180]. Various studies have shown that the potent oncogenicity of HER2Δ16 expression is attributed its ability to couple to and activate Src kinase [181,182]. Other mechanistic studies have shown that HER2Δ16 dimers activate several oncogenic signaling pathways, such as MAPK, PI3K/AKT, and FAK, that facilitate tumor cell proliferation and migration and promote EMT induction [183,184]. In addition to aberrant signaling, HER2Δ16 tumors have been reported to have distinct gene expression profiles with transcription factors associated with metastasis and cancer stem renewal being activated [185]. Studies show that ~89% of cases of lymph-node-positive disease progression in HER2-positive tumors involve the HER2 oncogenic variant HER2Δ16 [181]. Forced expression of HER2Δ16, but not wild-type HER2, promotes receptor dimerization, cell invasion, and trastuzumab resistance in cancer cell lines [181]. Therapeutic strategies are being studied to specifically target HER2Δ16 to overcome therapeutic resistance. For example, Huynh and Jones found that forces’ expression of microRNA-7 inhibited HER2Δ16-mediated breast tumorigenesis and reversed trastuzumab resistance in breast cancer cell lines [186].

### 3.6. HER2 Expression Heterogeneity, Stability, and Molecular Complexing in Tumors Limits Responsiveness to Trastuzumab

The differential expression of certain proteins can contribute towards the development of trastuzumab resistance. The heterogeneous expression of HER2 itself in the tumor has been implicated as a cause for resistance to anti-HER2 therapies [187]. A retrospective study conducted in 96 specimens from patients with invasive breast cancer with amplification of the *ERBB2* gene in the entire tissue section demonstrated there was intratumoral heterogeneity in the *ERBB2* gene amplification in a subset of patients. The presence of this heterogeneity correlated with reduced disease-free survival. The authors postulated that this could be due to genetic instability and this abnormal *ERRB2* amplification promotes cancer progression [188]. A subsequent study demonstrated that higher levels of heterogeneity are associated with poorer overall responses to trastuzumab [189]. One-third of patients with significant residual disease lose *ERBB2* amplification, and this change is associated with poor RFS. In the neoadjuvant therapy setting, Mittendorf et al. found that one-third of patients with residual tumor following treatment with trastuzumab-based therapy demonstrated loss of HER2 amplification [190]. Thus, tumor editing may be a major cause of resistance in many patients, suggesting follow-up HER2 assessments are required to tailor therapies.

The stability of HER2 protein in the tumor may also contribute to treatment resistance. The HSP90, for example, indirectly influences trastuzumab resistance by regulating HER2 protein stability. HER2:HSP90 complexes are involved in the formation of the appropriate conformation, stabilization, and activation of HER2 [191]. HER2 is sensitive to HSP90 inhibition, which results in faster degradation. Multiple studies have shown that inhibition of HSP90 restores sensitivity of HER2+ to trastuzumab [192,193,194,195]. In one recent example, Park et al. developed a new C-terminal HSP90 inhibitor, NCT-547. NCT-547 has significant anti-tumor activity when used in combination with trastuzumab in trastuzumab-resistant cell lines by promoting the degradation of both p95HER2 and full-length HER2 as well as disrupting the HER2 family members’ dimerization [194]. Importantly, it also significantly downregulates BCSC, as measured by reduced CD24^low^/CD44^high^ subpopulation, ALDH1 activity, and the formation of mammospheres. Clinical studies support the combination of trastuzumab and HSP90 inhibitors. A phase I clinical trial in patients with HER2+ trastuzumab refractory breast cancer found that combination of trastuzumab and tanespimycin showed appreciable anti-tumor activity [196]. A subsequent phase II clinical trial of 31 patients showed a 22% overall response rate, clinical benefit rate was achieved in 59% of the patients, the median progression-free survival was 6 months, and 17 months was the median overall survival [197]. Despite the impressive results, the use of HSP90 inhibitors remains underdeveloped due to several issues such as off target effects and poor solubility [198].

Numerous studies have showed that dopamine and cyclic AMP-regulated phosphoprotein (DARPP-32) contribute to trastuzumab resistance through forming complexes with HER2. DARPP-32 and its truncated form, t-DARPP, are expressed in most primary breast tumors, particularly in HER2+ tumors, and high levels of DARPP32 are associated with worse survival. DARPP-32 and t-DARPP proteins confer resistance to trastuzumab through sustained activation of the Akt pathway, which facilitates cell proliferation in trastuzumab resistance [199]. Belkhiri et al. showed that DARPP-32 and t-DARPP conferred resistance to trastuzumab by helping in the formation of protein complexes of HER2 with HSP90 [200]. Other studies suggest that t-DARPP facilitates the dimerization of IGF-1R with HER2 [149]. The results from these studies suggest that the expression of DARPP32 and t-DARPP32 may be acting as scaffolds enabling treatment resistance.

MUC4 is a glycoprotein that is often aberrantly overexpressed in breast cancer and associated with lymph node metastases [201]. In the trastuzumab-resistant breast cancer cell line, JIMT-1, MUC4 overexpression has been reported, by Nagi et al., to mask trastuzumab binding sites. Their data showed that this decreased binding was not due to intracellular retention, as another antibody displayed higher efficiency. The study proposed that the increased MUC4 expression concealed the trastuzumab-binding epitopes on HER2, and the masked epitopes were likely membrane-proximal. In addition to the trastuzumab binding epitopes, MUC4 disrupted the interactions between HER2 and its regular binding partners as well, causing activation of MAPK and PI3K signaling via alternate pathways [202]. A follow-up study by the same group showed, however, that despite intrinsic resistance in vitro, JIMT-1-derived xenograft tumors can be eliminated in vivo in xenograft models by ADCC (discussed below), suggesting that the therapeutic mechanisms of trastuzumab operate at varying densities of bound trastuzumab [203]. Despite that, Mercogliano et al. subsequently confirmed the earlier MUC4 finding and found that overexpression of MUC4 was linked to a poor response to adjuvant trastuzumab in HER2 breast cancer patients [204]. Furthermore, this study also revealed that TNF-α was a key driver of MUC4 overexpression. Other studies have shown that other glycoproteins may also contribute to masking HER2 in the tumor microenvironment, such as MUC1 and hyaluronan [205,206].

### 3.7. Immune-Mediated Mechanisms of Resistance

Resistance to trastuzumab may also be related to immune regulatory processes that are present at the time of treatment or develop following treatment. For example, upregulation of programmed death-ligand 1 (PD-L1) has been suggested as a mechanism of acquired trastuzumab resistance. Trastuzumab therapy of human HER2+ mouse tumors has shown that PD-L1 can be upregulated by stimulating IFNγ release [207]. PD-L1 suppresses immune effectors through the ligation of PD-1 [208]. Both NK cells and T cells are important sources of IFN-γ that could stimulate tumor PD-L1 expression [209]. However, whether PD-L1 has a major role is suppressing trastuzumab-induced immunity and clinical activity is not clear from clinic studies. The PANACEA (NCT02129556) phase Ib/II trial, in which patients were treated with a combination of trastuzumab and the anti-PD-1 antibody, pembrolizumab, showed only modest activity, with 6 of 40 PD-L1+ patients demonstrating an objective response [210].

Another immune-mediated mechanism of resistance is the suppression of natural killer (NK) cell antibody-dependent cellular cytotoxicity (ADCC). Darwich et al. reported that CHI3L1 expression was increased in sera obtained from trastuzumab-refractory patients compared to patients who responded to trastuzumab therapy and healthy controls. CHI3L1 impaired both NK-cell-mediated ADCC and cytotoxicity [211]. Mechanistically, the investigators found that CHI3L1 inhibited RAGE-mediated JNK signaling in NK cells, leading to defective polarization of the microtubule-organizing center (MTOC) and the prevention of degranulation. Murine modeling showed that CHI3L1 administration resulted in increased tumor growth and a change in the immune cell repertoire, including increased macrophages and decreased NK and T cell infiltration. The overexpression of CHI3L1 in the tumor cells impaired the efficacy of ADCC in mice models. Finally, it was also shown that a CHI3L1-neutralizing antibody synergized with trastuzumab to generate robust anti-tumor effects. 

A study investigating acquired resistance following prolonged exposure to trastuzumab increased the expression of 15 genes that contributed towards drug resistance [212]. A total of 5 out of these 15 genes belonged to the cytokine superfamily (CCL5, CXCL10, CXCL11, INFL1, and INFL2). Of these, the authors characterized CCL5 as a mediator of trastuzumab resistance through the activation of the ERK signaling pathway. The study also showed that increased expression of CCL5 in the tumor correlates with a lower pathological complete response (pCR) to trastuzumab following neoadjuvant therapy. CCL5 is also a potent chemoattractant of regulatory T cells (Tregs), which may also explain the lower pCR response following trastuzumab neoadjuvant therapy [213,214]. 

Lastly, tumors may resist trastuzumab therapy by upregulating complement regulatory proteins, particularly in the context of dual trastuzumab and pertuzumab therapy. Mamidi et al. showed that neutralization of the complement regulator proteins CD46, CD55, and CD59 enhanced the complement-mediated activities of trastuzumab and pertuzumab in HER2+ tumor cells [215]. While it is normal for human cells to be insensitive to human complement, tumors may upregulate them for increased evasion. Tsao showed that survival in HER2+ breast cancer is inversely related to tumor levels of CD55 and CD59 [104].

## 4. Resistance to Trastuzumab Drug Conjugates

In addition to its use unmodified, either alone or in combination, trastuzumab has been a foundation for a field of antibody drug conjugates (ADCs). ADCs utilize the specificity of the antibodies and the potency of the cytotoxic payloads [216]. Trastuzumab emtansine, also known as T-DM1, was one of the earliest trastuzumab-based ADCs to receive FDA approval in 2013 for the treatment of metastatic BC [216]. TDM-1 has been shown to be superior to trastuzumab-based therapies in various clinical settings [217,218]. The enhanced efficacy of T-DM1 over trastuzumab is attributed to the combined anti-tumor activity of both trastuzumab (e.g., inhibition of HER2-mediated signaling, inhibition of HER2 ECD shedding, and induction of ADCC) and DM1 (cell death due to mitotic catastrophe caused by the failure of formation of a functional mitotic spindle due to tubulin depolymerization) [219]. However, patients can be refractory and eventually progress on T-DM1 therapy, suggesting primary or adaptive resistance [220]. Like trastuzumab, resistance to T-DM1 can, theoretically, be mediated by reduced or loss of cell surface HER2 and/or glycoprotein masking (e.g., MUC4) [219]. This has been shown in both preclinical and clinical models. For example, relative to parental cell lines, decreased HER2 is seen in cell lines selected for TDM-1 resistance [221]. Two clinical studies, the KRISTINE trial and the ZEPHIR trial, assessed the influence of tumor heterogeneity on T-DM1 response [222,223]. Both the studies reported that, in patients with intratumor heterogeneity, where certain portions of the tumor had low HER2 expression, T-DM1 had limited benefits.

Deregulation of signaling pathways may also mediate T-DM1 resistance. A preclinical model demonstrated the loss of PTEN-induced T-DM1 resistance, which could be reversed using a PI3K inhibitor [224]. Similarly, defective Cyclin B1 induction by T-DM1 was reported to mediate acquired resistance in HER2+ BC cells [225]. Conversely, data from phase III TH3RESA study suggested no correlation between the efficacy of T-DM1 and PIK3CA mutation status in patient cohorts [226]. This indicates that there could be multiple mechanisms of resistance to T-DM1 and they could vary based on the models, clinical stage of the disease, and manner in which the drug was administered. Additionally, there are other plausible mechanisms that could mediate T-DM1 resistance, such as impaired trafficking of the HER2-T-DM1 complex, defective lysosomal degradation of T-DM1, high rate of HER2-T-DM1 recycling, and drug efflux proteins [45,219,224]. However, further investigations are required to understand their role(s) in T-DM1 resistance. As indicated by Hunter et al., there are likely to be significant differences in resistance mechanisms between trastuzumab and TDM-1, given the pre-eminence of the DM-1-mediated effects of the ADC [219].

Another trastuzumab-based drug conjugate, trastuzumab deruxtecan (DS-8201a or ENHERTU), a conjugate of trastuzumab and a topoisomerase-I inhibitor, deruxtecan, has also been approved for the treatment of breast cancer, including HER2-low breast cancer [227,228]. However, currently there is minimal literature on mechanisms of resistance to trastuzumab deruxtecan. It seems likely that they would overlap with those of T-DM1 and trastuzumab. 

## 5. Conclusions

Over the years, cumulative basic, translational, and clinical work has given insight into the multiple mechanisms of action and resistance to trastuzumab. As we increase our knowledge of trastuzumab and the interindividual genomic landscapes, trastuzumab regimens may also become more personalized, for example, by combination with neratinib in patients expressing higher levels of p95HER. The current standards of care in developed countries include combination with chemotherapy and or other drugs such as pertuzumab. Future treatments may involve combination with vaccine therapy, anti-ERBB2 tyrosine kinase inhibitors, antibody drug conjugates, and immune checkpoint inhibitors.

## Figures and Tables

**Figure 1 cancers-14-05115-f001:**
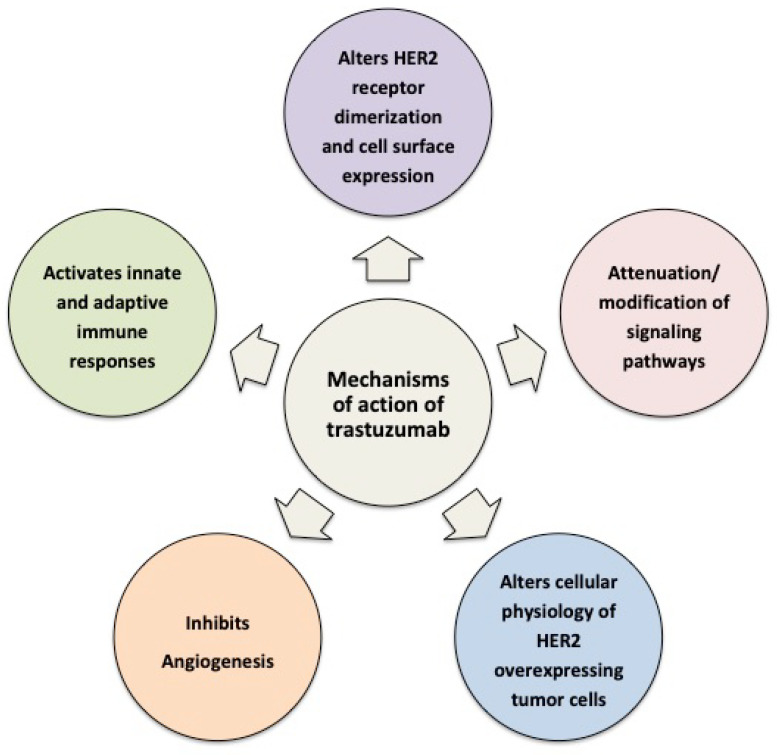
Schematic representation of the cellular and molecular mechanisms of action of trastuzumab.

**Figure 2 cancers-14-05115-f002:**
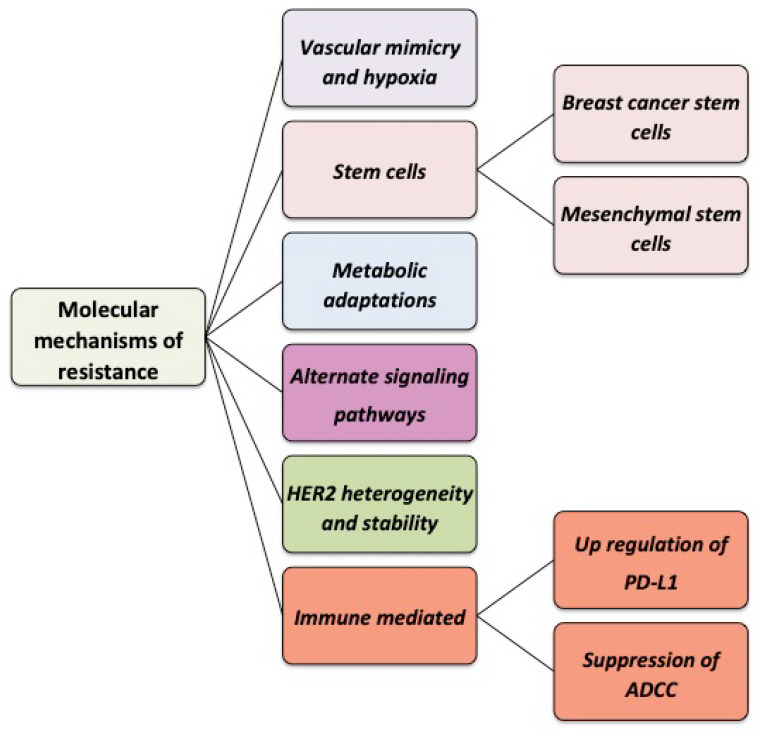
Schematic representation of the multiple mechanisms of resistance to trastuzumab.

## Data Availability

Not applicable.

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
