# Peer review of "Resistance to Trastuzumab"

_cancers, 2022, doi:10.3390/cancers14205115_

Round 1
Reviewer 1 Report
In this review, the authors give a comprehensive overview on the mechanism of action of trastuzumab as well as on the mechanisms behind the resistance to trastuzumab.
I think this review is well-written and interesting. I have some minor comments:
· Since trastuzumab is used in antibody-drug conjugates (it is not only a carrier in these drugs, rather trastuzumab is an active compound, i.g. it can evoke ADCC), I recommend discussing the trastuzumab-based antibody-drug conjugates in this manuscript.
· Maybe the authors can emphasize that some cancers have multiple mechanisms behind trastuzumab resistance.
· The authors mentioned that in the JIMT-1 trastuzumab resistant breast cancer cell line, the overexpression of MUC4 could mask trastuzumab binding sites leading to trastuzumab resistance. Other works from the same group showed that these in vitro trastuzumab-resistant cells were sensitive to trastuzumab-mediated ADCC in vivo, and trastuzumab could decrease the number of circulating and disseminated tumor cells despite trastuzumab resistance of the primary tumor. These works also distinguished between the mechanism of actions of trastuzumab through on its Fab and Fc part, and also distinguished between resistant mechanisms to trastuzumab through its Fab and Fc part.
Author Response
The authors thank the reviewer for the outstanding critique. We have modified the manuscript accordingly, as follows:
Comment 1: Since trastuzumab is used in antibody-drug conjugates (it is not only a carrier in these drugs, rather trastuzumab is an active compound, i.g. it can evoke ADCC), I recommend discussing the trastuzumab-based antibody-drug conjugates in this manuscript.
Response: We thank the author for the outstanding suggestion. We have added a section on page 16 regarding antibody drug conjugates.
Comment 2: Maybe the authors can emphasize that some cancers have multiple mechanisms behind trastuzumab resistance.
Response: The reviewer brings up a very good point. We have emphasized this aspect on page 2, second paragraph of the revised manuscript.
Comment 3: The authors mentioned that in the JIMT-1 trastuzumab resistant breast cancer cell line, the overexpression of MUC4 could mask trastuzumab binding sites leading to trastuzumab resistance. Other works from the same group showed that these in vitro trastuzumab-resistant cells were sensitive to trastuzumab-mediated ADCC in vivo, and trastuzumab could decrease the number of circulating and disseminated tumor cells despite trastuzumab resistance of the primary tumor. These works also distinguished between the mechanism of actions of trastuzumab through on its Fab and Fc part, and also distinguished between resistant mechanisms to trastuzumab through its Fab and Fc part.
Response: We thank the reviewer for pointing out this very interesting sequence of studies. We were able to find that study (Barok et al., 2007, Mol Cancer Ther) and have integrated in into the relevant section on page 14 of the revised manuscript.
Reviewer 2 Report
1. Page 3 of 25, HER2 cleaving metalloporteinases, such as ADAM17 and ADAM10….which can result in the generation of …p95HER2[15-17], the references 15 t didn’t mention anything regarding ADAM10 or ADAM17 relation to p95HER2. Could the authors point out the sentence in reference 16, 17 about such relationships?
2. “2C4” should be Pertuzumab and what is 4D5?
3. Page 3 of 25, the first paragraph are about ADAM10, ADAM17, and their shedding but the conclusion is strange. “ Overall, the observation support the idea that trastuzumab inhibits HER2 cleavage, thereby diminishing active unregulated membrane bond HER2 intracellular kinas domain.
4. Page 3 of 25, extracellular domain ECD, what is ECD”?
5. In section 3.2. the role of breast cancer stem cells in trastuzumab resistance. Page 10 of 25, the first paragraph is not related to breast stem cells.
6. In section 3.2 regarding breast cancer stem cells, relevant article could be found in “Cancer stem cell-targeted therapeutic approaches for overcoming trastuzumab resistance in HER2-positive breast cancer. Stem Cells. 2021 Sep;39(9):1125-1136.” This article also include table with treatments of breast stem cell.
7. Section 3.3 title, how to define metabolic adaptations?
8. Section 3.4 title, how to define “proximal signaling pathway?”
9. Page 8 of 25 last 3 sentence and the initial sentence of 9 of 25, the format is not the same.
Overall, this is an ambition article. Trying to gather all the currently available mechanisms. With segment studies statements make the reader hard to catch up.
Author Response
We thank the reviewer for the outstanding critique. We have modified the manuscript accordingly as follows:
Comment 1: Page 3 of 25, HER2 cleaving metalloporteinases, such as ADAM17 and ADAM10….which can result in the generation of …p95HER2[15-17], the references 15 t didn’t mention anything regarding ADAM10 or ADAM17 relation to p95HER2. Could the authors point out the sentence in reference 16, 17 about such relationships?
Response: We thank the reviewer for catching the error in this sentence. We reviewed that sentence and found that it was incorrectly written and referenced. We have revised the sentence to state “….ADAM10 and matrix metalloproteinases…” Additional references were added to supported this statement.
Comment 2: “2C4” should be Pertuzumab and what is 4D5?
Response: We have changed 2C4 to pertuzumab on page 3. We also added text on page 3 to further clarify that 4D5 is the parent murine antibody from which trastuzumab was derived.
Comment 3: Page 3 of 25, the first paragraph are about ADAM10, ADAM17, and their shedding but the conclusion is strange. “ Overall, the observation support the idea that trastuzumab inhibits HER2 cleavage, thereby diminishing active unregulated membrane bond HER2 intracellular kinas domain.
Response: We modified the text to enhance clarity to: “Overall, the observations support the idea that the effectiveness of trastuzumab could be due in part to its blockade of HER2 ectodomain cleavage which reduces production of oncogenic p95HER2.”
Comment 4: Page 3 of 25, extracellular domain ECD, what is ECD”?
Response: We noted that ECD is the abbreviation for the HER2 extracellular domain on Page 3.
Comment 5: In section 3.2. the role of breast cancer stem cells in trastuzumab resistance. Page 10 of 25, the first paragraph is not related to breast stem cells.
Response: We agree with the reviewer that gastric cancer stem cells are not related to breast cancer stem cells. However, our goal was to highlight where possible, other studies which support the overall hypothesis of stemness and trastuzumab resistance. We have made modifications to that paragraph and the subsequent paragraph to make this point clearer.
Comment 6: In section 3.2 regarding breast cancer stem cells, relevant article could be found in “Cancer stem cell-targeted therapeutic approaches for overcoming trastuzumab resistance in HER2-positive breast cancer. Stem Cells. 2021 Sep;39(9):1125-1136.” This article also include table with treatments of breast stem cell.
Response: We thank the author for pointing out this review article. We have cited it in the final paragraph of the relevant section on page 10.
Comment 7: Section 3.3 title, how to define metabolic adaptations?
Response: We have changed the title to that section to state “…metabolic changes” rather than "…metabolic adaptations”
Comment 8: Section 3.4 title, how to define “proximal signaling pathway?”
Response: We have removed the word proximal so that the title reads: “Activation of alternative signaling pathways”.
Comment 9: Page 8 of 25 last 3 sentence and the initial sentence of 9 of 25, the format is not the same.
Response: This has been corrected and thank the reviewer for identifying the inconsistency.
Reviewer 3 Report
This is a comprehensive and well-written review that would be of great interest to both clinicians and scientists. I recommend this manuscript for publication after minor changes.
Clinically, it is hard to distinguish if resistance to therapy in HER2-positive (breast) cancers is attributed only to trastuzumab (typically, trastuzumab with pertuzumab or trastuzumab-drug conjugates are administered with chemotherapy). The combination of HER2-targeted monoclonal antibody (or antibodies) and chemotherapy appears to be synergistic. This should be clarified.
Please define p85 and p110
HER2 clustering may be important in therapy response/resistance. In addition to homomers and HER1/HER3/HER4 heteromers, HER2 may also be forming heteromers with other receptor tyrosine kinases (which is mentioned later in the review). Perhaps include it upfront for clarity.
Please clarify notations and define receptors (e.g., IGF1R vs IGF-1R vs IGF-IR).
Beyond MUC4, overexpression of other bulky glycoproteins that could sterically inhibit the binding of trastuzumab to HER2 epitope may be relevant. This should be clarified as it may be an important mechanism of trastuzumab resistance.
Page 8, font changed.
Author Response
We thank the reviewer for the outstanding critique. We have modified the manuscript accordingly as follows:
Comment 1: Clinically, it is hard to distinguish if resistance to therapy in HER2-positive (breast) cancers is attributed only to trastuzumab (typically, trastuzumab with pertuzumab or trastuzumab-drug conjugates are administered with chemotherapy). The combination of HER2-targeted monoclonal antibody (or antibodies) and chemotherapy appears to be synergistic. This should be clarified.
Response: We thank the reviewer for pointing out the complexities of trastuzumab use in the clinic. We have revised the introductory paragraph to point out that understanding resistance of trastuzumab is confounded by combination use.
Comment 2: Please define p85 and p110
Response: We have defined these on page 4.
Comment 3: HER2 clustering may be important in therapy response/resistance. In addition to homomers and HER1/HER3/HER4 heteromers, HER2 may also be forming heteromers with other receptor tyrosine kinases (which is mentioned later in the review). Perhaps include it upfront for clarity.
Response: We have added text to point out that HER2 is also able to bind ot other partners on page 2.
Comment 4: Please clarify notations and define receptors (e.g., IGF1R vs IGF-1R vs IGF-IR).
Response: We thank the reviewer for pointing out the inconsistent naming and have gone through the document renaming all to IGF-1R. We have also defined it at first mention on page 2.
Comment 5: Beyond MUC4, overexpression of other bulky glycoproteins that could sterically inhibit the binding of trastuzumab to HER2 epitope may be relevant. This should be clarified as it may be an important mechanism of trastuzumab resistance.
Response: The reviewer has brought up an important point. We have included other examples, MUC1 and hyaluronan, in the relevant section (Page 15).
Comment 6: Page 8, font changed.
Response: We have corrected the font change on Page 8.